# Divergent combinations of *cis*-regulatory elements control the evolution of phenotypic plasticity

Mohannad Dardiry[1,2], Gabi Eberhard[1], Hanh Witte[1], Christian Rödelsperger[1], James W. Lightfoot[1,3], Ralf J. Sommer[1] *

1 Max-Planck Institute for Biology Tübingen, Tübingen, Germany, 2 Department of Genetics, Faculty of Agriculture, Cairo University, Giza, Egypt, 3 Max Planck Research Group Genetics of Behavior, Max Planck Institute for Neurobiology of Behavior–caesar, Bonn, Germany

* ralf.sommer@tuebingen.mpg.de

**Data Availability Statement:** RNA-seq data has been deposited at the European Nucleotide Archive under the study accession PRJEB59264 and

## Abstract

The widespread occurrence of phenotypic plasticity across all domains of life demonstrates its evolutionary significance. However, how plasticity itself evolves and how it contributes to evolution is poorly understood. Here, we investigate the predatory nematode *Pristionchus pacificus* with its feeding structure plasticity using recombinant-inbred-line and quantitative-trait-locus (QTL) analyses between natural isolates. We show that a single QTL at a core developmental gene controls the expression of the cannibalistic morph. This QTL is composed of several *cis*-regulatory elements. Through CRISPR/Cas-9 engineering, we identify copy number variation of potential transcription factor binding sites that interacts with a single intronic nucleotide polymorphism. Another intronic element eliminates gene expression altogether, mimicking knockouts of the locus. Comparisons of additional isolates further support the rapid evolution of these *cis*-regulatory elements. Finally, an independent QTL study reveals evidence for parallel evolution at the same locus. Thus, combinations of *cis*-regulatory elements shape plastic trait expression and control nematode cannibalism.

## Results and discussion

Resource polyphenisms are a special form of adaptive developmental plasticity that facilitate the exploitation of distinct food sources across animals [1]. Recent studies have started to identify the gene regulatory networks (GRNs) that control resource polyphenisms and other forms of plasticity [2,3]. One example is mouth-form plasticity in the hermaphroditic nematode *P. pacificus* with its predatory "eurystomatous" (Eu) and nonpredatory "stenostomatous" (St) morphs (Fig 1A and 1B) [4]. Mouth-form plasticity is controlled by the sulfatase EUD-1 that acts as a developmental switch: Expression above a certain threshold will result in the execution of the Eu form, whereas in the absence of *eud-1* expression, the St morph is formed [5]. *eud-1* is located in a multigene locus and is part of a complex GRN controlling *P. pacificus* mouth-form plasticity [6,7]. Knowledge about this GRN also provides a framework for natural variation studies and allows the testing of the contribution of plasticity for evolution.

PRJEB13695. All other data is available in the main text or the supplementary materials.

**Funding:** This work was funded by the Max-Planck Society through institutional funds to RJS, including the salaries of all co-authors. The funders had no role in study design, data collection and analysis, decision to publish, or preparation of the manuscript.

**Competing interests:** The authors have declared that no competing interests exist.

**Abbreviations:** CK, Coteau Kerveguen; CNV, copy number variation; Eu, eurystomatous; FPKM, Fragments Per Kilobase of transcript per Million mapped reads; GRN, gene regulatory network; LOD, Logarithm of the odds; NB, Nez du Boeuf; qRT-PCR, quantitative reverse transcription PCR; QTL, quantitative-trait-locus; RIL, recombinant-inbred-line; SNP, single nucleotide polymorphism; St, stenostomatous.

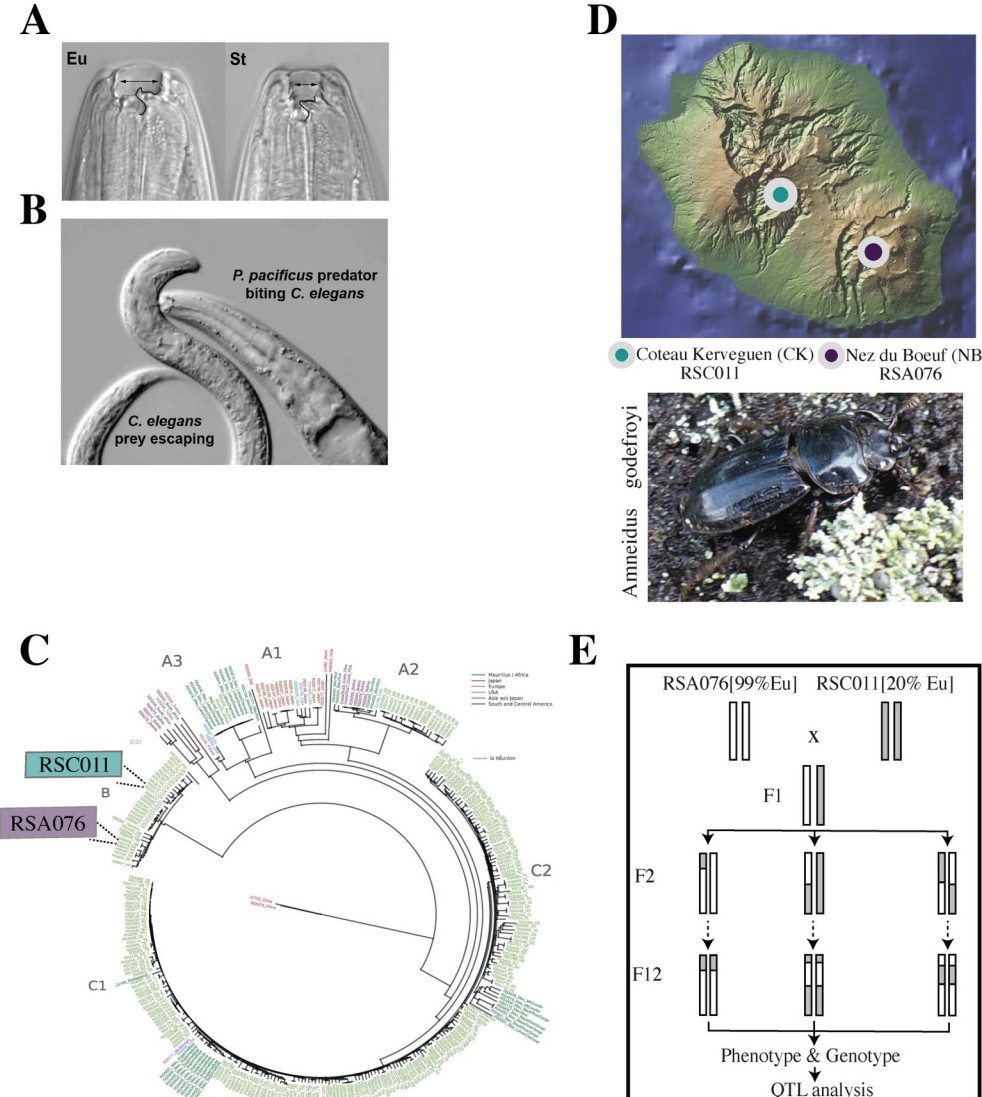

**Fig 1. Mouth-form plasticity and population structure of *P. pacificus*.** (**A**) *P. pacificus* mouth-form dimorphism. The predatory Eu form has a larger mouth opening and 2 teeth, in comparison to the St nonpredatory form with a narrow opening and a single tooth. (**B**) Killing behavior of an Eu adult biting *C. elegans* larval prey. (**C**) Phylogenetic relationship of a subset of the more than 300 *P. pacificus* isolates with strains from La Réunion Island indicated in green. Modified from [8]. Focal strains for subsequent analysis are RSA076 and RSC011 from clade B. (**D**) RSA076 and RSC011 are from neighboring high-altitude regions on La Réunion and were isolated from the endemic stag beetle *Amneidus godefroyi*, like all other clade B strains. The map is modified from [17]. Map figure made with GeoMapApp (www.geomapapp.org) / CC BY/CC BY [19]. (**E**) Crossing scheme of the 2 parental strains for RIL analysis. Eu, eurystomatous; QTL, quantitative-trait-locus; RIL, recombinant-inbred-line; St, stenostomatous.

## A single QTL regulates natural variation of mouth-form plasticity

To capture natural variation of mouth-form plasticity and the Eu versus St mouth-form ratio, we made use of a collection of around 1,500 *P. pacificus* isolates with more than 300 strains from La Réunion Island that were previously whole-genome sequenced (Fig 1C) [8]. We identified several pairs of closely related strains that differ in the preferential expression of mouth form when grown under standard laboratory conditions. For example, in *P. pacificus* clade B that is endemic to high-altitude locations on La Réunion, RSA076 from Nez du Boeuf (NB) is

nearly exclusively Eu, whereas the closely related strain *P. pacificus* RSC011 from Coteau Ker-veguen (CK) is preferentially St (80% St:20% Eu) (Fig 1D). Like all strains of clade B, RSA076 and RSC011 were isolated from the endemic stag beetle *Amneidus godefroyi*, which is restricted to high-altitude habitats like NB, CK, and neighboring regions (Fig 1D) [9].

We generated F1 hybrids between both strains and allowed F1 animals to self-fertilize for 12 generations to create 160 recombinant-inbred-lines (RILs) (Fig 1E). These RILs have different mouth-form ratios covering the complete range of 20% to 100% Eu, reflecting their mosaic homo-zygous genetic makeup (S1A Fig). We performed QTL analysis to statistically associate mouth-form ratios to genomic region(s) in the sequenced RILs (Fig 2A). This analysis initially identified 3 highly significant peaks across the genome. However, due to the genetic distance of RSA076 and RSC011 from the *P. pacificus* reference strain PS312, two of these signals were subsequently demon-strated to represent small X chromosome translocations [10] (S1B Fig). Thus, there is only 1 QTL, which spans a region of more than 200 kb. This QTL contains 35 predicted genes and, importantly, covers the previously described multigene switch locus including *eud-1* (Fig 2B). Besides the sulfa-tase-encoding *eud-1*, this locus contains the *eud-1* paralog *sul.2.2.1*, and 2 α-N-acetylglucosamini-dase-encoding genes (*nag-1* and *nag-2*), which result in all-Eu animals when mutated [6]. Thus, QTL analysis identified a single major locus regulating natural variation of mouth-form plasticity.

## Polymorphisms in *cis*-regulatory regions include copy number variations in potential transcription factor binding sites

Genetic variants at the multigene switch locus would be strong candidates to control mouth-form plasticity. We found a total of 41 single nucleotide polymorphisms (SNPs) within the 30-kb region spanning the multigene locus between RSA076 and RSC011. However, within the coding region of the four mouth-form associated genes, only a single nonsynonymous SNP was identified. This is found within *nag-2* and causes a Phe415Ile change. Using CRISPR/Cas-9 engineering, we introduced the RSA076 parental nucleotide into the RSC011 genetic background. Two independent lines carrying this substitution (*tu1489*, *tu1490*) did not show any change in the highly St phenotype, dismissing a role for this substitution in controlling mouth-form variation (S2 Fig and S1 Table). All other SNPs between the parental strains either are in intergenic or intronic regions or represent synonymous changes in genes of the multi-gene locus. Therefore, we focused on potential *cis*-regulatory variation as numerous studies have shown the involvement of *cis*-regulatory elements in adaptive divergence, particularly in promoter and enhancer regions of developmental control genes [11–16].

The highest number of SNPs between RSA076 and RSC011 are in the upstream region and the first intron of *eud-1* (S2 Table). Specifically, 5 SNPs in the upstream region and 1 SNP in intron 1 of *eud-1* are shared between related strains of RSA076 and RSC011 (Figs 2B and S3). In addition, we detected a 32-bp element that contains sequence similarity to a potential Forkhead transcrip-tion factor binding site (hereafter, Forkhead binding site (FBS)) in the upstream region of *eud-1* (Fig 2B). Interestingly, we observed copy number variation (CNV) of this 32-bp element between strains. RSA076 has 2 copies of this element, whereas RSC011 has only a single copy (Fig 2B). These SNPs and the CNV might be involved in the regulation of *eud-1*, as *eud-1* expression in RSA076 is 40% higher, consistent with its role in the specification of the Eu morph (Fig 2C).

## Systematic swapping experiments through CRISPR engineering identify *cis*-regulatory and intronic variants at the *eud-1* locus to control plasticity

To determine a potential role for the identified SNPs, we performed systematic swapping experiments using CRISPR/Cas-9 engineering. Specifically, we introduced substitutions in the Eu parental background RSA076 with the sequence variants of the RSC011 strain. However,

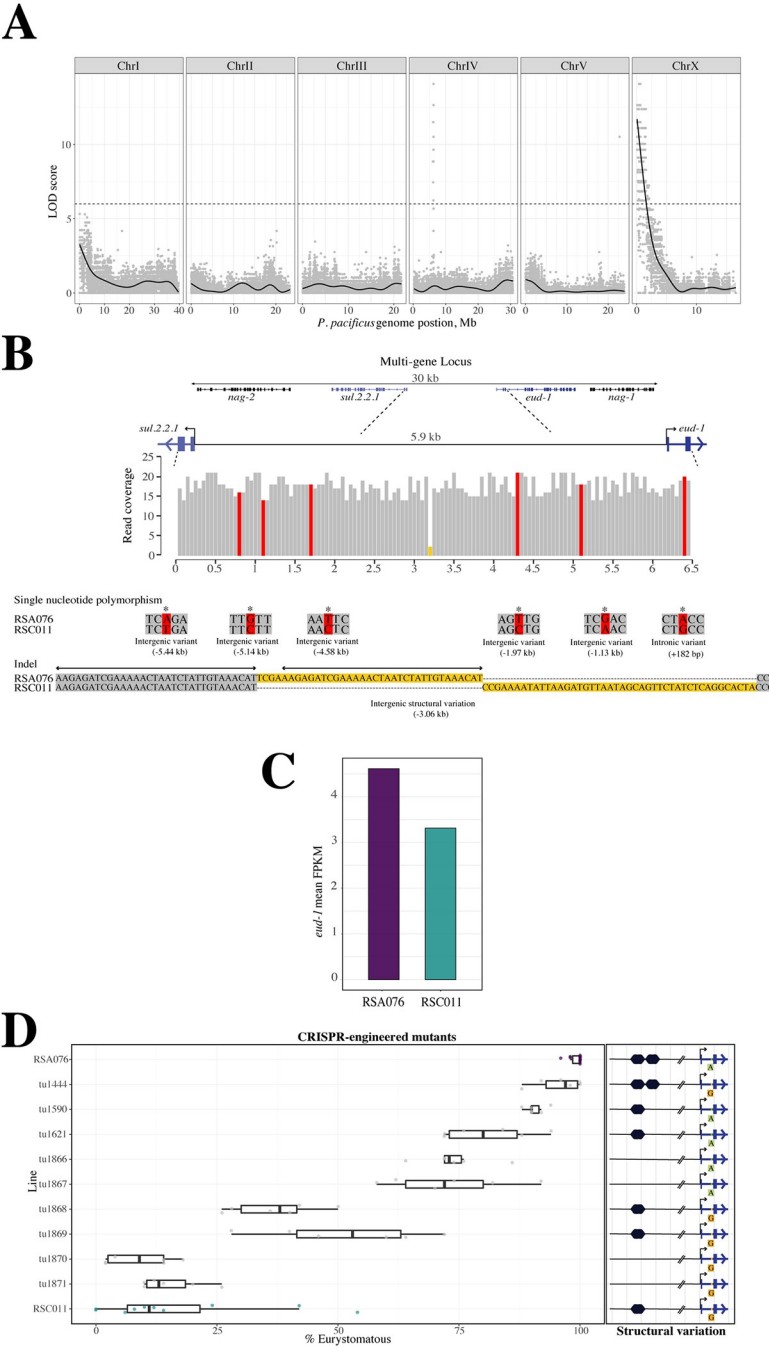

**Fig 2. RIL and QTL analyses between RSA076 and RSC011.** (**A**) QTL analysis reveals a single peak of around 200 kb at the left end of the X chromosome. (**B**) This region includes the multigene locus that contains the *eud-1* developmental switch gene. The upstream region of *eud-1* contains 5 SNPs (red) and 1 large CNV (yellow) with sequence differences indicated. An additional SNP exists in intron 1 (red). Arrows refer to the 32-bp repeated sequence, which contains the potential FBS (GTAAACAT). (**C**) RNAseq experiments indicate a 40% higher expression of *eud-1* in RSA076 relative to RSC011, consistent with its preferential Eu mouth-form. (**D**) Mouth-form ratios of various CRISPR-induced mutants introducing RSC011 variants in the RSA076 genetic background result in the sequential appearance of the RSC011 mouth-form ratio. Parental phenotypes are indicated in purple (RSA076) and green (RSC011), respectively. CNV of a potential forkhead transcription factor binding site in the *eud-1* promoter with RSA076 having 2 copies, whereas RSC011 has only a single copy (hexagonal shapes). Intron 1 has 1 single-nucleotide-polymorphism (G vs. A) between both strains. For detailed information, see S1 Data. CNV, copy number variation; FPKM, Fragments Per Kilobase of transcript per Million mapped reads; LOD, Logarithm of the odds; QTL, quantitative-trait-locus; RIL, recombinant-inbred-line; SNP, single nucleotide polymorphism.

replacing any of the SNPs did not change the mouth-form ratio in the resulting lines (Figs 2D, S4, and S5 and S3 Table). In contrast, manipulating CNV of the FBS revealed strong changes in mouth-form ratios. When we deleted one of the 2 copies of the FBS in RSA076, both resulting lines (*tu1590*, *tu1621*) showed a reduction of the predatory morph to 75% to 90% Eu animals (Fig 2D). Elimination of the second copy (*tu1866*, *tu1867*) resulted in a further reduction of the Eu form (Fig 2D). Strikingly, however, if we swapped the A-G SNP in intron 1 in the presence of deletions of the FBS, we observed even more drastic changes in mouth-form ratios. First, the RSA076(*tu1590*) allele that harbors only 1 copy of the FBS showed a strong reduction of the Eu mouth form (40% to 55% Eu) after introducing the A-G swap (*tu1868*, *tu1869* in Fig 2D). Second, when we introduced the A-G swap in a line that has both FBS copies deleted (*tu1870*, *tu1871*), the mouth-form ratio is below 20% Eu, similar to RSC011 animals (Fig 2D). Thus, CNV of the FBS in the upstream region of *eud-1* regulates mouth-form plasticity synergistically with a single SNP in the first intron, and different combinations of these regulatory elements can alter the mouth-form ratio between 10% and 100% Eu (Fig 2D and S4 Table). Importantly, the mutant lines *tu1868* and *tu1869*, while closely mimicking the RSC011 mouth-form ratio, do not completely phenocopy this strain. As such, this finding may indicate the involvement of other strain-specific background effects in the manifestation of the phenotype.

Sequence comparisons of the *eud-1 cis*-regulatory region in a broader diversity of *P. pacificus* strains provided further support for the rapid evolution of both identified elements (S6 Fig). Notably, strains of other *P. pacificus* clades, including the "wild-type" PS312, have 3 FBS copies and are also preferentially Eu (S6 Fig). Additionally, further sequence alignment of the FBS element revealed the existence of a similar sequence in the first intron of *eud-1* (Fig 3A). This additional element is located in the middle of intron 1 but is identical in the RSA076 and RSC011 strains. We used CRISPR/Cas-9 technology to manipulate this element. While small deletions have little to no effect on mouth-form plasticity, a 4-bp insertion already shifts the mouth form substantially towards the St morph (Fig 3B). Subsequently, we were able to generate 2 independent 31-bp deletions that completely eliminate the sequence homologous to FBS. Both of these lines show 0% Eu animals, and all worms develop the St morph (*tu1905*, *tu1906*) (Fig 3B and S5 Table). Thus, the first intron of *eud-1* contains an additional regulatory element that when eliminated results in an all-St phenotype similar to *eud-1* knockouts [5]. This intronic element shows no sequence variation between wild isolates of *P. pacificus*, but strong sequence divergence in the sister species *P. exspectatus*, which is strongly St (S7 Fig) [17,18].

### *Cis*-regulatory variants affect *eud-1* expression

Next, we used quantitative reverse transcription PCR (qRT-PCR) experiments to provide direct evidence that the engineered CRISPR lines shifting the mouth-form ratio from Eu to St are indeed affecting *eud-1* expression. For this, we measured *eud-1* expression relative to the normalized *eud-1* expression of the parental RSA076 strain. We found that lines with the deletion of both FBS and the intronic swap (*tu1870*, *tu1871*) show strongly reduced *eud-1* expression, similar to RSC011 (Fig 3C). This was further validated in the line with the 31-bp deletion in intron 1, which also resulted in a strong reduction of *eud-1* expression (Fig 3C). These findings indicate that the *cis*-regulatory variants in natural isolates of *P. pacificus* affect *eud-1* expression and thereby influence mouth-form execution.

### Population-scale whole-genome sequencing suggests the Eu pattern to be ancestral

Finally, we tested if the QTL shows any evidence for selection and wanted to determine the direction of evolutionary change in mouth-form ratio. For that, we employed available

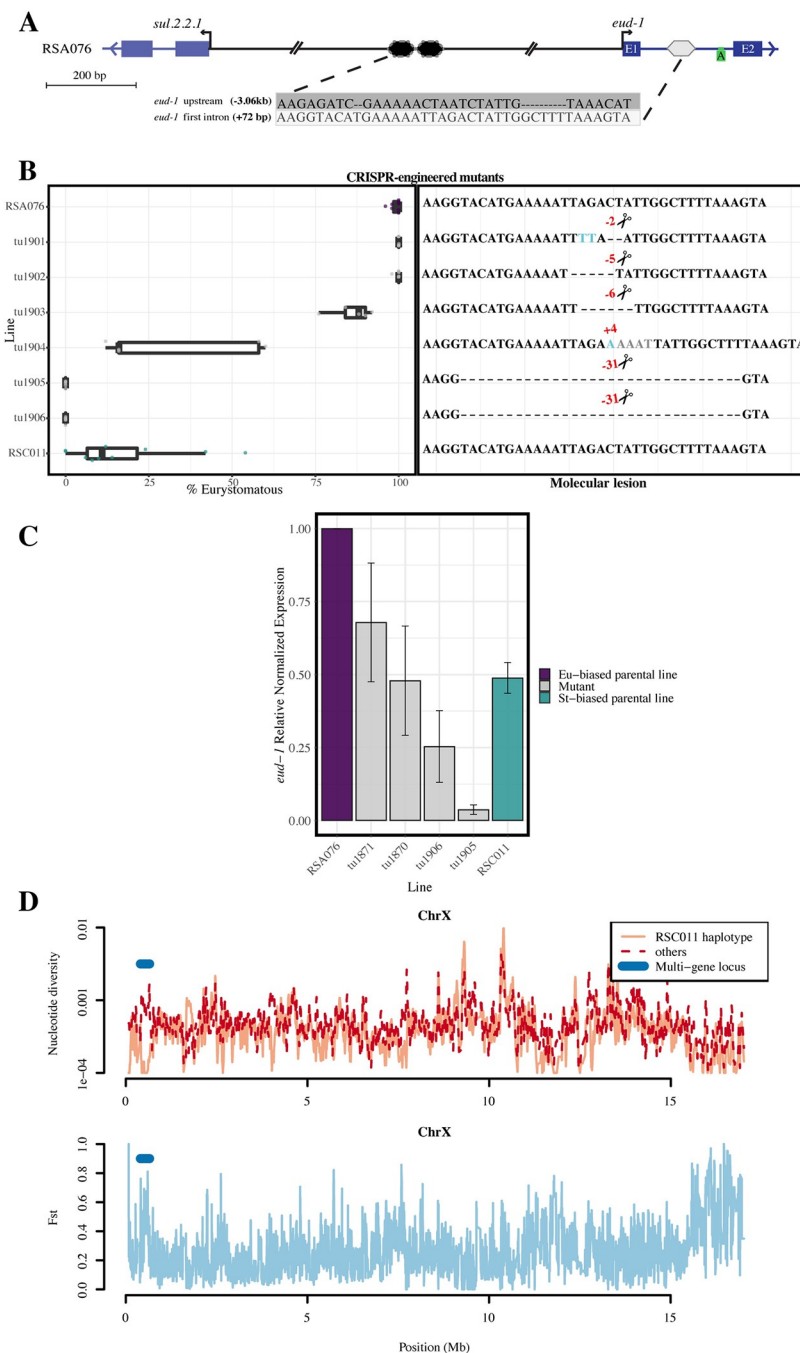

**Fig 3. Functional analysis through CRISPR/Cas-9 engineering and evolutionary divergence of *eud-1 cis*-regulatory elements. (A)** Sequence comparison of the transcription factor binding site in the *eud-1* promoter reveals a related sequence in intron 1. This sequence element is identical between RSA076 and RSC011. **(B)** Mouth-form ratios of various CRISPR-induced mutants introducing deletions of this intronic element in the RSA076 genetic background. A complete deletion results in a 100% St (0% Eu) phenotype. Parental phenotypes are indicated in purple (RSA076) and green (RSC011), respectively. **(C)** Quantitative PCR experiments of selected mutant strains exhibit lower *eud-1* expression correlating with the preferential St mouth form. **(D)** Nucleotide diversity and F_st data of the strains from CK and NB based on population-scale whole-genome sequencing. CK strains with the RSC011 haplotype show low diversity at the QTL peak on the X chromosome. Nucleotide diversity of the other chromosomes are shown in S8 Fig. For detailed information, see S2 Data. CK, Coteau Kerveguen; Eu, eurystomatous; NB, Nez du Boeuf; QTL, quantitative-trait-locus; St, stenostomatous.

population-scale whole-genome sequencing data for the strains used in this study [8] to compare their genotype with mouth-form ratios (S8 Fig). Specifically, we compared all 10 available strains from CK, where RSC011 was isolated, with representative strains from NB, the origin of RSA076 (S8 Fig). Strikingly, we identified 4 of the 10 strains from CK that had a preferentially St mouth form and shared the same haplotype at the *eud-1* locus with RSC011 (S8A Fig). In contrast, the remaining isolates exhibited either the RSA076-type variants at the *eud-1* locus or a mixed pattern (RSC010, RSC173) and were all preferentially Eu (S8B Fig). No such variation was seen in the strains from NB with all strains being preferentially Eu and having the RSA076 haplotype (S8B Fig). These results indicate that the RSA076 Eu pattern is ancestral in *P. pacificus* clade B and that the RSC011 St phenotype has recently evolved. Moreover, the CK strains with the RSC011 haplotype at the QTL peak showed a remarkably low diversity suggesting that this haplotype was introduced very recently into the population (Figs 3D and S8C). To distinguish if these patterns are shaped by neutrality or natural selection will require future sampling efforts with a higher temporal resolution.

### Parallel evolution shapes natural variation of mouth-form plasticity in *P. pacificus*

Given the central role of the *eud-1* and its neighboring genes in the mouth-form decision [5,6], we wondered if this large locus represents a hotspot for natural variation in phenotypic plasticity. Thus, we tested for evidence of parallel evolution at this locus by repeating RIL and QTL analysis using 2 more distantly related *P. pacificus* strains. Specifically, we performed an RIL experiment between the highly Eu PS312 strain from California, from which the reference genome is derived, and the highly St strain RSB020 that belongs to a different clade [8]. Indeed, our analysis again identified a single major QTL on the X chromosome that covers the *eud-1* locus (S9 Fig). Given the large phylogenetic distance between PS312 and RSB020, many more SNPs and other variants are found in the *eud-1* regulatory region. Taken together, independent RIL and QTL analyses of genetically diverse strains indicate a major role for the multigene locus and *eud-1* in natural variation of mouth-form plasticity.

With these studies, we have combined RIL- and QTL-based population genetic analysis with CRISPR-mediated experimentation to study mouth-form plasticity evolution. Our observations strongly support a role for distinct combinations of *cis*-regulatory elements at a single developmental switch gene controlling mouth-form plasticity. Through these studies, we (i) demonstrate the existence of natural variation associated with phenotypic plasticity; (ii) identify its molecular architecture; and (iii) establish population differentiation coupled to this regulatory mechanism.

## Supporting information

**S1 Text. Supplementary materials and methods.**
(DOCX)

**S1 Fig. Recombinant inbred lines (RILs). (A)** Mouth-form ratios of a subset of the RILs representative of the full range of phenotypes. **(B)** Highly significant markers of ChrV and ChrIV display similar inheritance patterns as ChrX markers. Theses similarities indicate genetic linkage between the markers, which, in turn, implies that all these markers must be on the same chromosome. The x-axis displays the 55 RILs, and the y-axis indicates chromosomal markers. For detailed information, see S3 Data.
(DOCX)

**S2 Fig. Nonsynonymous variant swap in *nag-2*.** Mutant lines *tu1489* and *tu1490* were generated by introducing the RSA076 variant in the RSC011 background, i.e., swapping (A) with (T). Mouth-form score was counted for 3 replicates each line. Statistical analysis shows no significant difference between the mutant lines and RSC011. For detailed information, see S4 Data.
(DOCX)

**S3 Fig. Selection of candidate variants in *eud-1* first intron and upstream promoter.** Detection of 7 candidate variants in the *cis*-regulatory region of the switch gene *eud-1*. Candidates are categorized as follows: 1 intronic SNP (green box), 5 intergenic SNPs (black box), 1 deletion representing the CNV (red box). Candidates were selected by sequence comparison between the Eu-biased parental line (RSA076) against 3 closely related St-biased lines (RSC011, RSC012, RSC008), while RSC011 represents the St-biased parental line. Gene identifier RSB001000006991, top-left, represents the gene *sul.2.2.1*. Gene identifier RSB001000004808, top-right, represents the gene *eud-1*.
(DOCX)

**S4 Fig. Intergenic variants swap in 2 candidate positions (−1.13 kb and −1.97 kb).** Mutant lines *tu1485* and *tu1487* represent the (−1.13 kb) candidate, while *tu1504* and *tu1505* represent the (−1.97 kb) candidate. Swapping variants aimed at shifting the mouth-form score from Eu-biased to St-biased. Mouth-form score was counted for 3 replicates each line. For detailed information, see S5 Data.
(DOCX)

**S5 Fig. Intronic variant effect (+182 b). (A)** Mutants with downstream deletion to the intronic swap variant; *tu1445*, *tu1446*, *tu1444* display weaker mouth-form change than the mutant with the targeted swap; *tu1444*. Both *tu1447* and *tu1444* harbor the same 4-bp deletion. For detailed information, see S6 Data. **(B)** Sequence alignment of the mutants. The nucleotide under the star represents the targeted swapped nucleotide. Dashes represent deletions.
(DOCX)

**S6 Fig. Evolutionary divergence of the *eud-1* upstream and intronic region in a worldwide diversity of *P. pacificus* strains.** Several strains including the wild-type PS312 from California have 3 copies of the transcription factor binding site and contain the intronic "A" polymorphism similar to RSA076. Strain color code: green: clade B; orange: clade A; blue: clade C and light purple: outgroup. For detailed information, see S7 Data.
(DOCX)

**S7 Fig. Intronic sequence evolution.** Intronic sequence highly similar to the 32-nucleotide block in the *eud-1* promoter, (+72 bp), display sequence conservation across *P. pacificus* clades. RSA076 represents clade B, PS312 represents clade A, and RSB020 represents clade C. The same intronic sequence show divergence when compared to the most closely related *Pristionchus* species (*P. exspectatus* and *P. arcanus*). Dashes represent deletions, and red boxes represent nucleotide variation.
(DOCX)

**S8 Fig. Evidence for selection and direction of evolutionary change between RSA076 and RSC011.** (**A**) Extended phylogeny of clade B strains of *P. pacificus*. Note that only 10 strains are available from CK, one of the most remote places on La Réunion Island. Only a subset of the many strains of NB and CC are shown. (**B**) Mouth-form divergence and natural variation at the *eud-1* locus of all 10 CK-derived and 10 selected NB-derived strains. Only 4 CK strains share the RSC011 haplotype, whereas the majority of the others strains have the RSA076

haplotype consistent with RSC011 representing the derived character. **(C)** Nucleotide diversity and $F_{st}$ data of the 20 strains from CK and NB based on population-scale whole-genome sequencing. CK strains with the RSC011 haplotype were compared to NB and CK strains with the RSA076 haplotype. For detailed information, see S8 Data.
(DOCX)

**S9 Fig. QTL analysis between PS312 and RSB020.** QTL analysis of 94 RILs reveals a single peak of around 900 kb at the left end of the X chromosome. For detailed information, see S9 Data.
(DOCX)

**S1 Table. *nag-2* mutant lines.** $N$ = 150, 3 replicates ($n$ = 50) for all lines. % Eu, percent eurystomatous animals; n.a., not applicable. Genomic position in relation to RSB001 reference genome.
(DOCX)

**S2 Table. SNPs distribution within the 30-kb multigene locus.** Highest number of intergenic SNPs was detected in the intergenic region between *eud-1* and *sul.2.2.1*.
(DOCX)

**S3 Table. Mutant lines for candidate positions (−1.13 kb and −1.97 kb).** $N$ = 150, 3 replicates ($n$ = 50) for all lines. % Eu, percent eurystomatous animals; n.a., not applicable. Genomic position in relation to RSB001 reference genome.
(DOCX)

**S4 Table. Mutant lines for causative variants.** $N$ = 300, 6 replicates ($n$ = 50) for mutants and $N$ = 500, 10 replicates ($n$ = 50) for parental lines; % Eu, percent eurystomatous animals; n.a., not applicable. Genomic position in relation to RSB001 reference genome.
(DOCX)

**S5 Table. Mutant lines for intronic region (+72 bp).** $N$ = 250, 5 replicates ($n$ = 50) for mutants and $N$ = 500, 10 replicates ($n$ = 50) for parental lines; % Eu, percent eurystomatous animals; n.a., not applicable. Genomic position in relation to RSB001 reference genome.
(DOCX)

**S1 Data. Fig 2 detailed data.** Representing the QTL data of RILs generated between RSA076 and RSC011, J3 animals RNA-seq data, and causative mutants' mouth-form score in the RSA076 background.
(XLSX)

**S2 Data. Fig 3 detailed data.** Representing intronic mutants's mouth-form score in the RSA076 background, qPCR results, and population genetics data.
(XLSX)

**S3 Data. S1 Fig detailed data.** Representing mouth-form score of RILs generated between RSA076 and RSC011, besides markers segregation patterns of ChrIV as similar to ChrX.
(XLSX)

**S4 Data. S2 Fig detailed description of the *nag-2* mutants' mouth-form score in the RSC011 background.**
(XLSX)

**S5 Data. S4 Fig detailed description of the intergenic variants swap within the 2 candidate positions (−1.13 kb and −1.97 kb) in the RSA76 background.**
(XLSX)

**S6 Data. S5 Fig detailed description of the eud-*1* intronic mutants in the RSA076 background.**
(XLSX)

**S7 Data. S6 Fig detailed description of 29 *P. pacificus* natural isolates mouth-form score.**
(XLSX)

**S8 Data. S8 Fig detailed data.** Representing mouth-form score of *P. pacificus* natural isolated from 2 localities in clade B, NB and CK, besides population genetics data.
(XLSX)

**S9 Data. S9 Fig QTL data of the RILs generated between PS312 and RSB020.**
(XLSX)

## Acknowledgments

We thank Drs. M. Herrmann, J. Rochat, and the La Réunion field team for species and strain isolation. The authors thank members of the Sommer lab for discussion.

## Author Contributions

**Conceptualization:** Mohannad Dardiry, Ralf J. Sommer.

**Data curation:** Mohannad Dardiry, Christian Rödelsperger, James W. Lightfoot.

**Formal analysis:** Mohannad Dardiry, Christian Rödelsperger, Ralf J. Sommer.

**Funding acquisition:** Ralf J. Sommer.

**Investigation:** Gabi Eberhard, Hanh Witte, James W. Lightfoot, Ralf J. Sommer.

**Methodology:** Christian Rödelsperger, James W. Lightfoot.

**Supervision:** Ralf J. Sommer.

**Visualization:** Mohannad Dardiry, Christian Rödelsperger.

**Writing – original draft:** Mohannad Dardiry, Christian Rödelsperger, James W. Lightfoot, Ralf J. Sommer.

**Writing – review & editing:** James W. Lightfoot, Ralf J. Sommer.

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
