## [Editor Report · Decision Letter 0]

30 Mar 2023

Dear Dr Sommer, 

Thank you for submitting your manuscript entitled "Divergent combinations of cis-regulatory elements control the evolution of phenotypic plasticity" for consideration as a Short Report by PLOS Biology.

Your manuscript has now been evaluated by the PLOS Biology editorial staff as well as by an academic editor with relevant expertise and I am writing to let you know that we would like to send your submission out for external peer review.

Once your full submission is complete, your paper will undergo a series of checks in preparation for peer review. After your manuscript has passed the checks it will be sent out for review. To provide the metadata for your submission, please Login to Editorial Manager (https://www.editorialmanager.com/pbiology) within two working days, i.e. by Apr 03 2023 11:59PM.

Kind regards,

Ines

--

Ines Alvarez-Garcia, PhD

Senior Editor

PLOS Biology

---

## [Decision Letter · Decision Letter 1]

6 Jun 2023

Dear Dr Sommer,

Thank you for your patience while we considered the revised version of your manuscript entitled "Divergent combinations of cis-regulatory elements control the evolution of phenotypic plasticity" for publication as a Short Report at PLOS Biology. The manuscript has been evaluated by the PLOS Biology editors, the Academic Editor and by two reviewers.

Based on the reviews, we are likely to accept this manuscript for publication, provided you satisfactorily address the remaining points raised by Reviewer 2. Please also make sure to address the data and other policy-related requests stated below.

We expect to receive your revised manuscript within two weeks. 

*Published Peer Review History*

*Press*

Sincerely,

Ines

--

Ines Alvarez-Garcia, PhD

Senior Editor

PLOS Biology

Fig. 2A, D, E; Fig. 3B-D; Fig. S1A, B; Fig. S2; Fig. S4; Fig. S5A; Fig. S6; Fig. S8B, C and Fig. S9

BLURB

Please also provide a blurb which (if accepted) will be included in our weekly and monthly Electronic Table of Contents, sent out to readers of PLOS Biology, and may be used to promote your article in social media. The blurb should be about 30-40 words long and is subject to editorial changes. It should, without exaggeration, entice people to read your manuscript. It should not be redundant with the title and should not contain acronyms or abbreviations. For examples, view our author guidelines: https://journals.plos.org/plosbiology/s/revising-your-manuscript#loc-blurb

FINANCIAL DISCLOSURE

Please include grant numbers and the URLs of any funder's website and use initials to identify authors who received the funding. Also, please describe the role of any sponsors or funders in the study design, data collection and analysis, decision to publish, or preparation of the manuscript. If the funders had no role in any of the above, include this sentence at the end of your statement: "The funders had no role in study design, data collection and analysis, decision to publish, or preparation of the manuscript.

Reviewers' comments

Rev. 1:

This is a superb investigation of the genetic basis of variation in phenotypic plasticity. The feeding structure polyphenism in Pristionchus pacificus is iconic due to the well-documented plasticity and the extensive genetic, natural history and phylogenetic work on this species.

This study compares closely related lineages from Reunion that differ in their constitutive proportions of eurystomatous (predatory) and stenostomatous forms. The authors take a variety of approaches to understand the source of this difference.

1. Hybrids between the 2 forms were used to create RILs that show a relatively continuous variation of mouth form proportions from strictly 'eurystomatous' to strictly 'stenostomatous'.

2. Examination of candidate loci responsible for the difference in proportions reveals a single important QTL on the X-chromosome. This QTL is in the region that contains eud-1 previously identified as important for the plastic response. The eurystomatous line has 2 copies of an intergenic region compared to a single copy in the stenostomatous line. The copy number sequence and the intron sequence are quite similar.

3. Utilizing CRISPR/Cas9, a variety of deletions of the copy number variant and the intron in eud-1 are created. Small deletions have little effect. A 4 bp addition induces a major shift from eurystomatous towards stenostomatous. And deleting the4 whole sequence produces individuals that are strictly stenostomatous.

4. Population genetic analysis indicates a recent selective sweep in the eurystomatous population, which appears to be the derived condition.

5. Assessment of worldwide natural variation reveals that populations carry from 1 to 3 upstream elements and variation in the intronic region as well.

Remarkable for identifying genetic elements directly conditioning plasticity and demonstrating how variation in these elements can create variation in expression of the alternate mouth forms.

It will be interesting to see further work that investigates how these variants respond to the natural cues inducing eurystomaty.

Although I have read previous work by this group, I am not expert in Pristionchus or nematodes, but I found the paper to be very clearly written.

Rev. 2:

In this interesting paper, Dardiry et al. describe their investigation into the genetic basis of natural variation in the expression of a polyphenic trait: mouth form in Pristionchus pacificus nematodes. The authors present evidence implicating allelic variation in the eud-1 sulfatase in determining the tendency to make alternative mouth forms at different ratios. Overall, the study integrates genetic variation and the expression of phenotypic plasticity. That is, they identify natural variation at the genetic level that strongly impacts the expression of a plastic phenotype that has important stochastic and environmental inputs.

The methods and results are cutting-edge, combining meticulous phenotyping and genotyping with CRISPR-mediated perturbations. Though there are a few mysteries, their results go a long way towards defining the key sequence variants that underlie their quantitative trait loci (QTL). I do have some concerns, comments, and suggestions, however, that I believe will strengthen the paper:

Page 3, par. 2:

Figure 1E's F12 diagrams don't reflect the homozygosity expected from selfing for this long.

Page 4, bottom par:

It is interesting that simultaneous deletion of one copy of the upstream repeated element and conversion of the eud-1 intron 1 SNP in RSA076, which largely converts the QTL to the RSC011 genotype, is not sufficient to perfectly mimic the RSC011 phenotype of 20% Eu formation. Only when both repeats are deleted do we get to the RSC011 level. This should be discussed at some point. One also wonders whether, in such a background, other QTL might be identifiable that also contribute to the all-Eu state of RSC076.

Page 6, par. 1 and Pages 14-15 (Figure 3 and associated text):

Regarding a possible selective sweep of the RSC011 QTL allele: This is one possibility, but I am skeptical. It would seem that once we specify any specific, recently evolved haplotype among a larger population set, we expect the diversity to fall compared to the whole set. Further, that the RSC011 haplotype is not fixed even in the Clade B worms that live in the spatially confined Nez de Boeuf A. godefroyi-associated habitat in which both RSC011 and RSA076 live actually suggests the opposite: a tendency to form the Eu mouthform is not under strong selection, and thus a recently evolved, mild loss-of-function allele in eud-1 is basically neutral. Overall, a claim or suggestion of a selective sweep seems premature.

Pages 12-13:

Figure 2 needs some help to be easily interpreted. A few specific issues first:

* In panel B, the label "P. pacificus genome scale, Mb" is not helpful. It appears to be copied over from the bottom of panel A, but here it is clear kilobases (Kb), not Mb.

* The numbered scale on the read coverage plot has some digits out of order (from 3-4).

The biggest issue: I struggled to reconcile the lower part of panel B with panel C, but finally figured it out: In panel B the first, conserved repeat copy in 076 is in gray at the left, while the second is part of the strain-specific yellow sequence. It would be more clear to get rid of panel C, and modify panel B to better indicate the repeated part. For example, an arrow indicating the extent of each copy over the alignment would be very clear. The end result will have the same information, but take less space and be easier to comprehend.

---

## [Editor Report · Decision Letter 2]

22 Jul 2023

Dear Dr Sommer,

Thank you for the submission of your revised Short Report entitled "Divergent combinations of cis-regulatory elements control the evolution of phenotypic plasticity" for publication in PLOS Biology. On behalf of my colleagues and the Academic Editor, Laurence Hurst, I delighted to let you know that we can in principle accept your manuscript for publication, provided you address any remaining formatting and reporting issues. These will be detailed in an email you should receive within 2-3 business days from our colleagues in the journal operations team; no action is required from you until then. Please note that we will not be able to formally accept your manuscript and schedule it for publication until you have completed any requested changes.

PRESS

Sincerely, 

Ines

--

Ines Alvarez-Garcia, PhD

Senior Editor

PLOS Biology
